# Medial entorhinal cortex activates in a traveling wave in the rat

J Jesús Hernández-Pérez*, Keiland W Cooper, Ehren L Newman

Department of Psychological and Brain Sciences, Indiana University Bloomington, Bloomington, United States

**Abstract** Traveling waves are hypothesized to support the long-range coordination of anatomically distributed circuits. Whether separate strongly interacting circuits exhibit traveling waves remains unknown. The hippocampus exhibits traveling 'theta' waves and interacts strongly with the medial entorhinal cortex (MEC). To determine whether the MEC also activates in a traveling wave, we performed extracellular recordings of local field potentials (LFP) and multi-unit activity along the MEC. These recordings revealed progressive phase shifts in activity, indicating that the MEC also activates in a traveling wave. Variation in theta waveform along the region, generated by gradients in local physiology, contributed to the observed phase shifts. Removing waveform-related phase shifts left significant residual phase shifts. The residual phase shifts covaried with theta frequency in a manner consistent with those generated by weakly coupled oscillators. These results show that the coordination of anatomically distributed circuits could be enabled by traveling waves but reveal heterogeneity in the mechanisms generating those waves.

## Introduction

The recent recognition that numerous cortical circuits exhibit traveling waves when engaged motivates the hypothesis that traveling waves are a basic organizing principle of cortical activity (*Muller et al., 2018*). By this hypothesis, traveling waves generate macroscopic activation dynamics that coordinate interaction between anatomically distributed circuits. It remains to be shown, however, whether anatomically distributed circuits generate the coordinated activation dynamics using traveling waves as predicted by this hypothesis. To date, traveling waves have only been observed in isolated circuits (*van Ede et al., 2015*). The hippocampus, for example, has been established to activate in a traveling wave (*Lubenov and Siapas, 2009*; *Patel et al., 2012*; *Zhang and Jacobs, 2015*). It is unknown, however, whether the entorhinal cortex, an anatomically separated area but one that is highly inter-connected with the hippocampus, also activates in a traveling wave. A goal of this work was to test whether the entorhinal cortex, like the hippocampus, activates in a traveling wave.

Hippocampal activity exhibits traveling waves that have been hypothesized to contribute to the functionality of the circuit. An ~8 Hz 'theta' rhythm is readily observed at individual electrodes (*Grastyán et al., 1959*; *Kahana et al., 1999*) and comparison of theta activity across electrodes reveals progressive theta phase shifts between the dorsal/septal and the ventral/temporal poles of the hippocampus in humans and rats alike (*Lubenov and Siapas, 2009*; *Patel et al., 2012*; *Zhang and Jacobs, 2015*). Theta activity is well studied in terms of its underlying physiology (for reviews, see *Buzsáki, 2002*; *Dickson et al., 2000*; *Vinogradova, 1995*). Functionally, theta activity is attributed with supporting navigation (*Blair et al., 2007*; *Brandon et al., 2011*; *Burgess et al., 2007*; *Hasselmo et al., 2009*; *Newman and Hasselmo, 2014*; *Onslow et al., 2014*), mediating associational learning (*Caplan et al., 2003*; *Hasselmo et al., 2002*; *Hernández-Pérez et al., 2015*; *Hernández-Pérez et al., 2016*; *Honey et al., 2018*; *Muller et al., 2018*; *Norman et al., 2006*; *Patel et al., 2012*), and coordinating the distinct components of the entorhinal-hippocampal circuit

*For correspondence:
jjhp25@gmail.com

**Competing interests:** The authors declare that no competing interests exist.

(*Mizuseki et al., 2009*). These proposed functions are not mutually exclusive. Yet, for each function, entorhinal input to the hippocampus must arrive at a specific theta phase locally within the hippocampus. Because theta phase varies along the hippocampus, we hypothesize that theta phase also varies along the entorhinal cortex to form a traveling wave.

The medial entorhinal cortex (MEC), like the hippocampus, exhibits clear extracellular field theta rhythms at individual electrodes (*Mitchell and Ranck, 1980*). Simultaneous recordings of theta activity in dorsal MEC and dorsal hippocampus reveal a stable phase offset between these areas in terms of theta phase (*Mitchell and Ranck, 1980*; *Mizuseki et al., 2009*), consistent with the idea that theta activity facilitates interaction between these areas. Recordings performed across the layers of MEC at a single position along the dorsal-ventral axis show that theta activity is largely synchronous in layers III-V and reverses phase between layers III and I (*Mitchell and Ranck, 1980*; *Alonso and García-Austt, 1987*; *Chrobak and Buzsáki, 1998*; *Mizuseki et al., 2009*; *Quilichini et al., 2010*). It is unknown, however, whether theta phase varies systematically along the dorsal-ventral axis of the MEC, reflecting the existence of a traveling wave.

Traveling waves, as we use this term here, are spatiotemporal oscillations with steady phase shifts as a function of space, consistent with the previous use of this term to describe phase offsets in the hippocampus (*Lubenov and Siapas, 2009*; *Patel et al., 2012*; *Zhang and Jacobs, 2015*). Multiple mechanisms are capable of generating steady phase shifts, including variable waveform asymmetries, propagating pulses in an excitable network, delayed excitations from a single oscillator, and weakly coupled oscillators (*Cole and Voytek, 2017*; *Ermentrout and Kleinfeld, 2001*). Weakly coupled oscillators have been shown to be more likely to underlie hippocampal traveling theta waves than propagating pulses or delayed excitations (*Zhang and Jacobs, 2015*). The contribution of waveform asymmetry to hippocampal traveling theta waves, however, has not been explicitly explored.

Here, we aimed to establish whether traveling waves exist in the MEC and to identify the nature of the underlying mechanism. To do so, we used custom high-density electrode arrays to record the field potential at regular intervals along the dorsal-ventral axis of the MEC in freely behaving rats. Our recordings revealed reliable phase shifts across the dorsal-ventral extent of the MEC, indicating the existence of traveling waves. Further analyses of these recordings revealed that much of the phase shift was generated by systematic variation in waveform asymmetry and that the residual shifts resembled those generated by weakly coupled oscillators.

## Results

### Theta phase shifts gradually across the dorsal-ventral axis of the MEC

To determine whether there is a traveling wave in the MEC, we used custom high-density electrode arrays to record extracellular field potentials at regular intervals along the dorsoventral axis of MEC while the rats completed laps on a circle track for food rewards (*Figure 1*). In each of the six rats implanted, gradual phase shifts in the ~8 Hz theta-band were apparent in the raw recordings (representative example shown in *Figure 1C*).

Dorsal sites were phase advanced relative to ventral, as can be seen in the cycle-triggered averages of the raw LFP, with the phase shifting at an average rate of 26.36 °/mm ± 2.64 °/mm (*Figure 2A and D*). This mean shift was significantly different from zero (rank sum [max]=21 [21], p=0.03, n=6). The observed offsets were fit well by a linear Pearson correlation ($r^2$ = 0.7627), indicating that the phase changed steadily across the sampled length of the MEC (*Figure 2D*). The phase estimates for these analyses were derived via a Hilbert transform to match the methods used previously to identify hippocampal traveling waves (*Lubenov and Siapas, 2009*; *Patel et al., 2012*; *Zhang and Jacobs, 2015*). From these observations, we conclude that there are theta-band traveling waves in the MEC.

Theta activity was coordinated along the dorsal-ventral axis, as indicated by analyses of phase locking and coherence. The phase offset between any two channels was relatively stable over the duration of the recording (*Figure 2B and E*) given the median phase locking observed between sites of 0.90 [0.85–0.95]. The strength of phase locking (i.e., mean resultant length) decreased modestly but reliably as a function of the distance between the electrodes (slope of −0.067/mm ± 0.011; rank sum [max]=0 [21], p=0.03, n=6; *Figure 2E*). Theta coherence across the MEC was 0.73 [0.66–0.81].

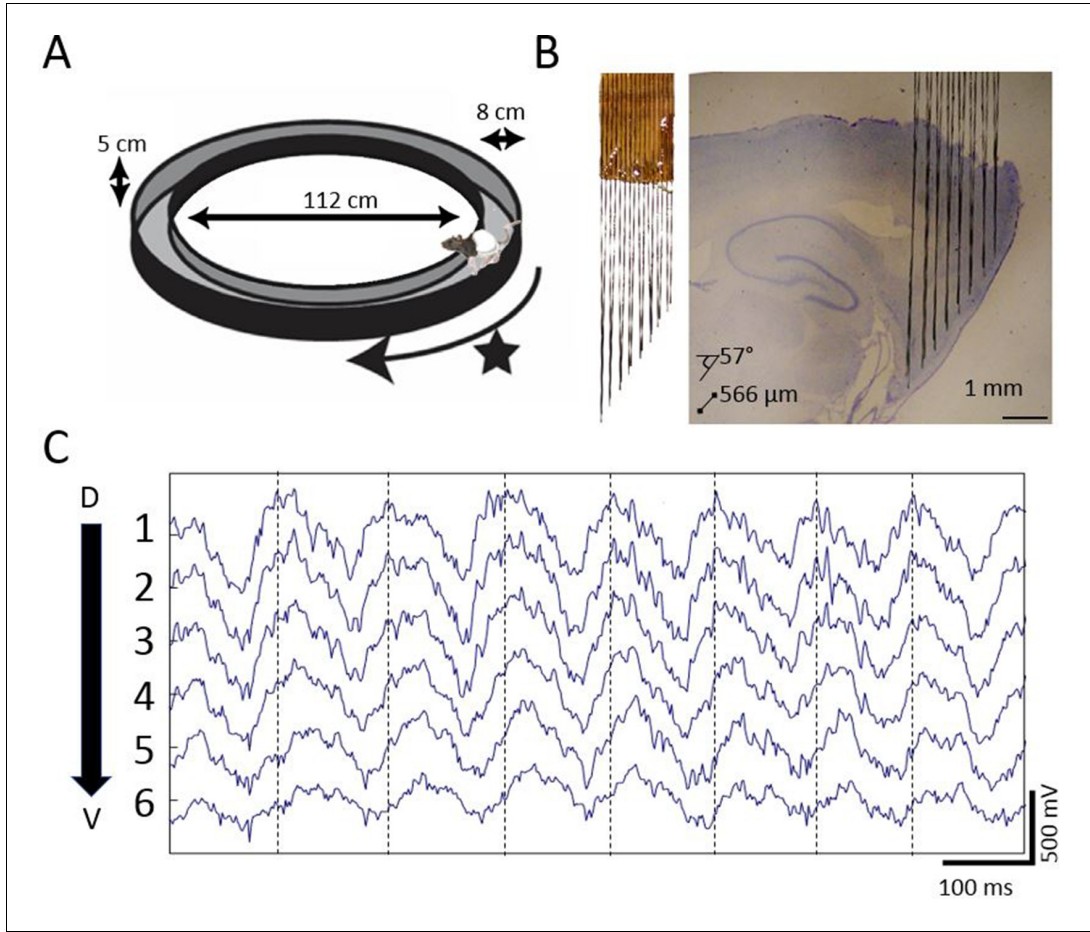

**Figure 1.** Extracellular field potential recordings along ~3 mm of the dorsal-ventral axis of the MEC in behaving rats reveals a gradual phase shift of theta. (**A**) Rats completed laps on a circular track for food rewards delivered at a fixed location marked by the star. (**B**) Custom electrode arrays with fixed interelectrode spacing (566 µm and 57° orientation relative to horizontal plane) were used to sample the field potential along ~3 mm of the long axis of the MEC. Array depth was controlled by a micro-drive (not shown). (**C**) Example broadband LFP traces from adjacent electrodes in MEC reveal regular phase shifts. Vertical lines mark the theta peaks from the dorsal-most channel to facilitate visual comparison of phase shifts across channels.

The online version of this article includes the following figure supplement(s) for figure 1:

**Figure supplement 1.** Histological micrographs for the six rats implanted for this study, marking estimated electrode placement at the time of sacrifice (orange and blue stars) and inferred positions at the time of the recording analyzed and described in the main text (green squares).

Coherence also decreased as a function of the distance between electrode pairs (slope: 0.067/mm ± 0.018; rank sum [max]=21 [21], p=0.03, n=6; *Figure 2C and F*). These patterns are consistent with previously reported shifts in theta phase and coherence across the long axis of the hippocampus (*Lubenov and Siapas, 2009*; *Patel et al., 2012*). We observed no significance change in theta power along the long axis of the MEC (slope of 0.73 [–0.08 – 0.93] AU/mm; rank sum [max]=19 [21], p=0.09, n=6; *Figure 2G*).

To evaluate the possibility that the observed phase shifts could be attributed to variable placement of the electrode tips with respect to the cortical layers of MEC, we implanted one rat with a variant of the electrode array that sampled both along the length of the MEC and across its layers (*Figure 3*). This enabled comparison of theta phase differences along versus across the layers of MEC. The positions of individual electrodes were fixed relative to those of other electrodes in the arrays, so that the displacement for each electrode pair with respect to axes running parallel to (i.e., along) the layers of MEC and perpendicular to (i.e., across) the layers was known. For each electrode

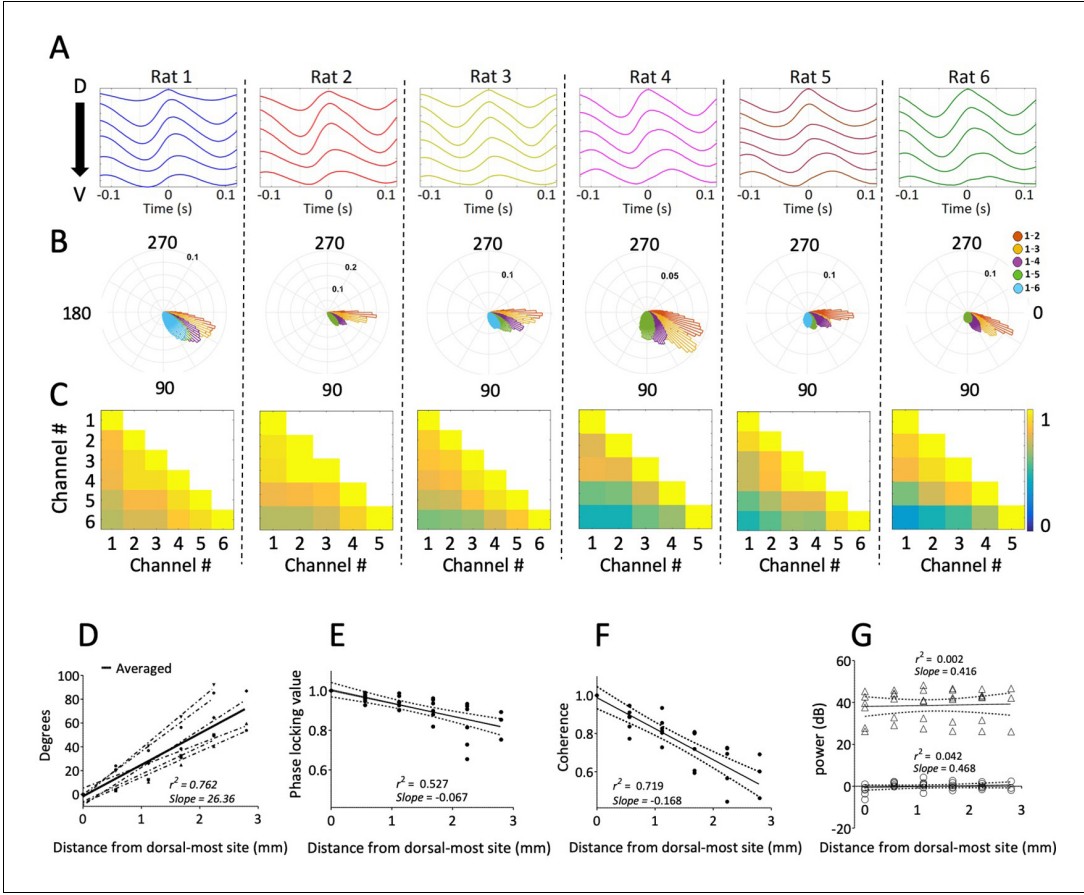

**Figure 2.** Consistent gradual phase shifts were observed along the long axis of MEC in all animals. (**A**) Average theta waves across channels, computed as an event-related average of the raw LFP triggered on the peak of theta for the dorsal-most channel, show similar theta phase shifts across channels for each rat. (**B**) Histogram of theta phase differences for each electrode relative to the dorsal-most electrode for each rat. (**C**) Theta-band coherence between all electrode pairs for each rat. Electrode 1 corresponds to the most dorsal position. (**D**) Summary of theta phase shifts of each electrode relative to the dorsal-most electrode as a function of the distance between the electrodes. See also *Figure 3*. (**E**) Phase locking (mean resultant length) between each electrode and the dorsal-most electrode, plotted as a function of inter-electrode distance, shows consistently high phase-locking across the dorsal-ventral axis. (**F**) Summary of theta-band coherence changes between each electrode and the dorsal-most electrode (corresponding to left-most column of the matrices shown in C) plotted as a function of inter-electrode distance. (**G**) Spectral power of theta as a function of distance from the dorsal-most-site. Triangles indicate raw power. Circles indicate power relative to each animal's across-electrode mean power. Solid lines in panels (E), (F), and (G) reflect regression trendlines across rats, and the dashed lines reflect 95% confidence intervals on the regression coefficients.

pair, we plotted the observed theta phase shift as a function of the displacement along one axis or the other (*Figure 3C–F*). Phase shift was positively correlated with displacement 'along the cell layer' (rho = 0.58) but not with displacement 'across the layers' (rho = 0.02). From this, we conclude that the phase shifts observed along the dorsal-ventral axis were not due to variable placement across the layers of MEC. The lack of evidence of clear phase shifts resulting from variable placement of the electrodes across the layers of the MEC is consistent with prior reports that there are no phase off-sets across layers III–V (*Mitchell and Ranck, 1980*; *Alonso and García-Austt, 1987*; *Chrobak and Buzsáki, 1998*; *Mizuseki et al., 2009*; *Quilichini et al., 2010*).

Next, we sought to verify that the phase shifts observed in the field potential were indicative of the local neural activity. To do so, the electrodes were moved from layer III, where theta is clearly visible in the LFP but units are sparse, to layer II, where units are densely packed to facilitate recording of multi-unit activity (*Figure 4A*). We then asked whether the timing of multi-unit activity mirrored

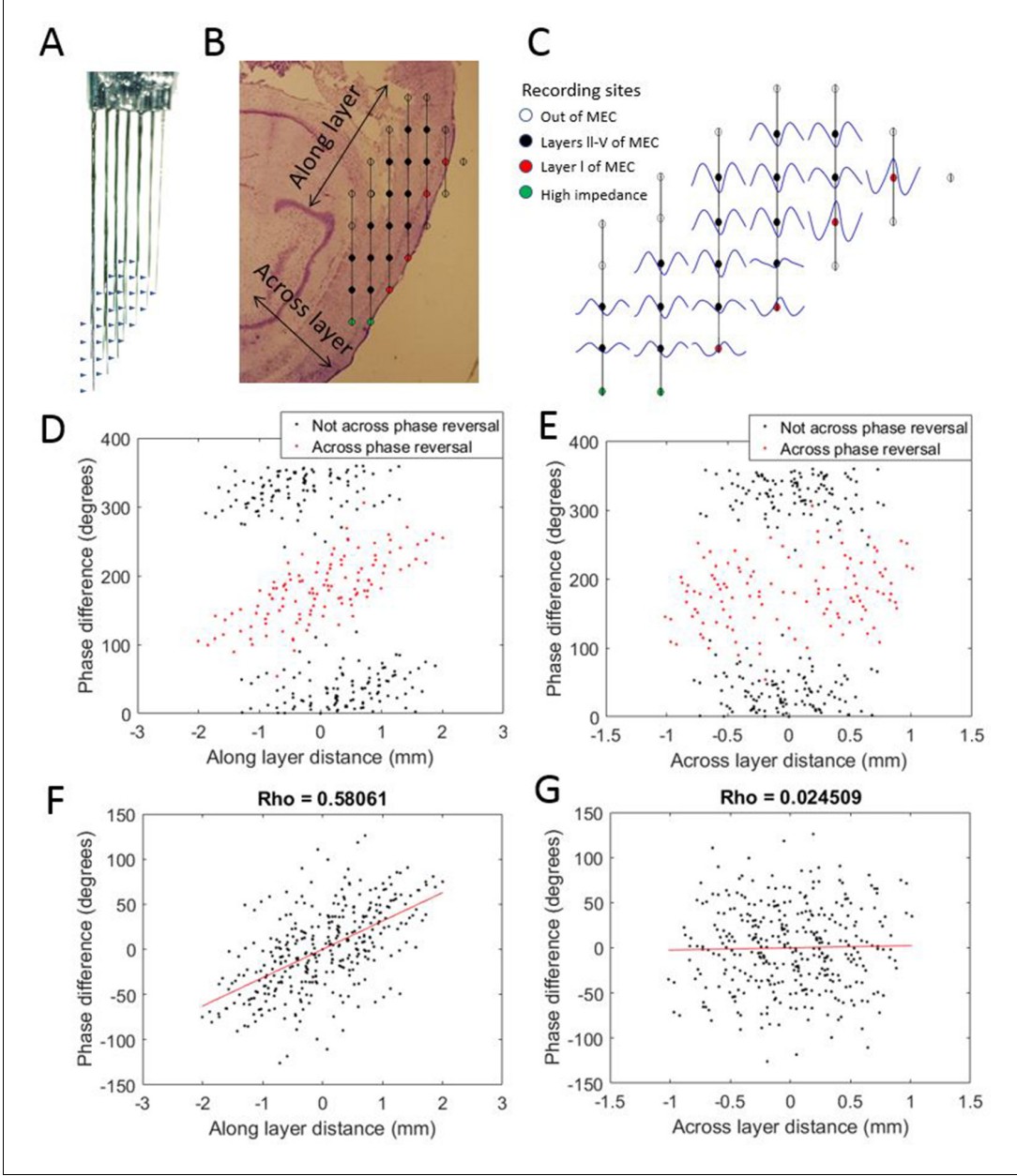

**Figure 3.** Progressive theta phase shift expressed along layers but not across layers in MEC. (**A**) Custom multi-shank electrode array with tips positioned to allow simultaneous recording along and across the cortical layers of the MEC. Arrow heads indicate the positions of recording sites on the adjacent shank. (**B**) Reconstructed position of the electrode array based on histology, probe geometry, and electrophysiological markers. Color coding of individual sites indicates whether they were excluded for either being beyond the bounds of the MEC (white) or having high impedance (green), and whether they were found to have a phase reversal (red) relative to the remaining sites (black). (**C**) Cycle-average theta waves for all electrodes, triggered off of peaks on the most dorsal electrode of layer l. The progressive theta phase shift is observed along the same layer, but not across the layer, note the phase inversion of 180° between layer I and layers III–V. (**D**) The theta phase difference for each electrode pair plotted as a function of the distance between the corresponding electrodes along the cortical layers (as shown in panel [B]) reveals a bimodal distribution that separated cleanly based upon whether the electrode pair spans the phase reversal expected between layers II and I (red) or not (black). (**E**) Same as panel (D) but phase differences were plotted as a function of distance across the cortical layers (as shown in panel B). (**F**) Applying a 180° offset to the red dots shown in panel (D) collapses the multi-modal distribution into a single modal distribution with a strong positive correlation between distance along the cortical layer and phase difference. (**G**) Applying a 180° offset to the red dots shown in panel (E) also collapses the multi-modal distribution

*Figure 3 continued on next page*

*Figure 3 continued*

into a single modal distribution but with no notable correlation between distance across layers and phase difference.

that of the field potential. In three animals, clear multiunit activity was observable simultaneously across four or more of the channels, enabling us to test for reliable delays between sites. The multiunit activity exhibited clear theta-rhythmicity as expected (*Mitchell and Ranck, 1980*). The amplitude envelope of the 600–3000 Hz spiking-band activity showed a progressive phase shift across channels, resembling that which we observed in the theta field potential (*Figure 4B*). Computing the cross-correlogram of the amplitude envelope between each channel and the dorsal-most channel showed that multiunit activity also generated a traveling wave with a lag consistent with a 33.20°/ mm ± 2.69°/mm phase shift (*Figure 4C and D*). This phase shift was significantly greater than zero (rank sum [max]=21 [21], p=0.03, n=3) and was fit well by a linear trendline ($r^2$ = 0.9269). The slope

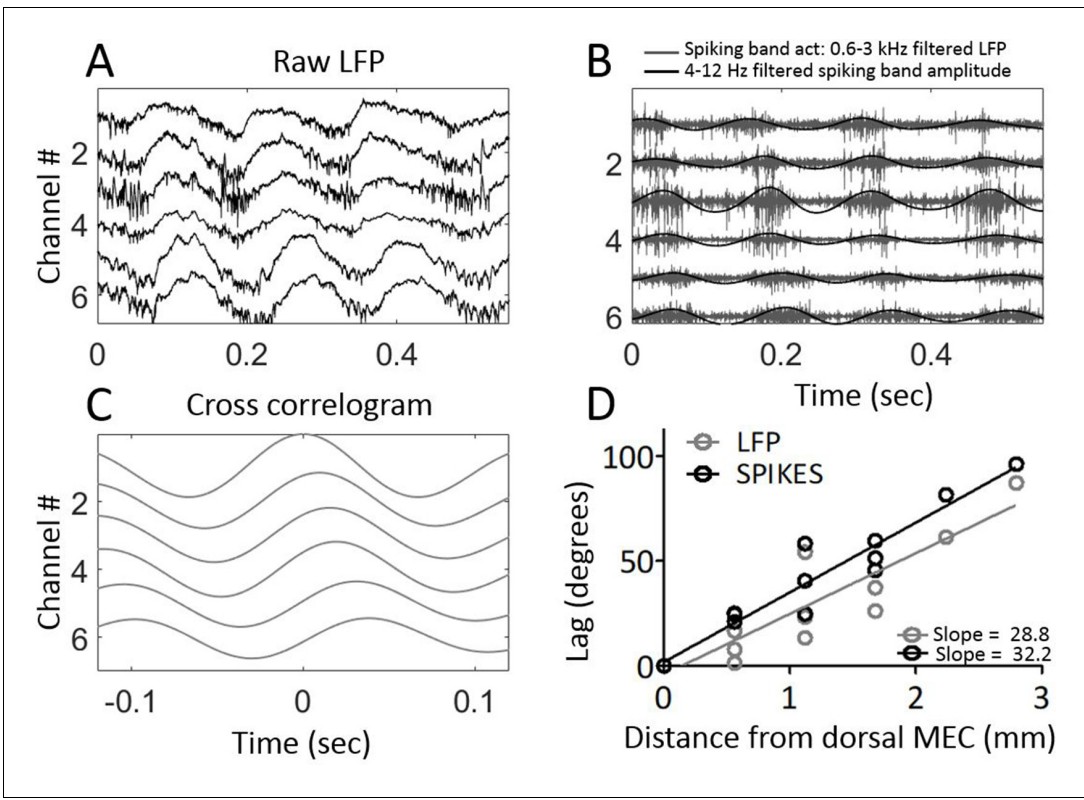

**Figure 4.** Multi-unit activity exhibited traveling waves along the dorsal-ventral axis. (A) Representative example of multi-unit activity locked to the trough of the local theta that was observable in the broadband LFP when the electrode arrays were lowered into MEC layer II. (B) Bandpass filtering the broadband signal to 600–3000 Hz to focus on the 'spiking-band,' shown in gray, shows that the timing of these theta-rhythmic fluctuations in multi-unit activity varied across the dorsal-ventral axis of the MEC in this representative example. Low-pass filtering of the spiking-band amplitude, shown in black, highlights the theta rhythmic structure of the spiking activity. (C) Cross-correlations of the spiking-band amplitude between the dorsal-most channel and each of the other channels from the same trial, as shown in panels (A) and (B), resemble the traveling LFP theta waves. (D) A consistent pattern of increasing lags was observed across the three rats that had multiunit activity on four or more channels simultaneously. The lag to the cross-correlation peaks (black points and best-fit line) increases as a function of distance from the dorsal-most electrode (n=3 animals with four or more electrodes spiking simultaneously). The slope closely mirrors the slope in LFP theta phase offsets (gray points and best-fit line from the same three animals). The linearly increasing lags confirm that multi-unit activity activates at progressively longer delays along the dorsal-ventral axis.

relating lag to position was not significantly different between theta-rhythmic multiunit activity and LFP theta phase across these three animals (rank sum [max]=0 [6], p=0.25, n=3).

## Theta waveform varies along the dorsal-ventral axis of the MEC

Visual inspection of the raw LFP traces suggested that the waveform of theta varied along the dorsal-ventral axis (*Figure 1C*). To analyze these changes, we compared the duration of the rising phase to the duration of the decaying (falling) phase, and the duration of the peak to the duration of the trough, using a variant of the waveform asymmetry index (*Belluscio et al., 2012*; *Cole and Voytek, 2017*). Analysis of this asymmetry demonstrated that theta was asymmetric at dorsal sites and significantly less so at relatively ventral sites, whether analyzed in terms of the asymmetric durations of the rise and decay (−0.43 [−0.57 – 0.35] vs. −0.11 [−0.13 – 0.09]; rank sum [max]=0 [21], p=0.03, n=6) or in terms of the asymmetric duration of the peak and trough durations (0.21 [0.00–0.29] vs. −0.10 [–0.16 – 0.01]; rank sum [max]=0 [21], p=0.03, n=6) as shown in *Figure 5*. In other words, at dorsal sites, the rising phase was significantly shorter than the decaying phase (signed rank [max]=0 [21], p=0.03, n=6) and the peaks were significantly longer than the troughs (signed rank [max]=20 [21], p=0.06, n=6). At ~2.8 mm from the dorsal edge of the MEC, the rising phase was still significantly shorter than the decaying phase (signed rank [max]=20 [21], p=0.06, n=6) but the duration of the peaks was not significantly different than that of the troughs (signed rank [max]=3 [21], p=0.16, n=6).

A consequence of varying waveform is that the time required for a theta wave to travel from dorsal sites to ventral sites (i.e., conduction delay) varied as a function of theta phase. Such phase-dependent differences are washed out when theta phase is estimated by filtering the signal down to the theta-band and applying the Hilbert transform. The Hilbert transform was used in the preceding analyses to match the methods used previously in order to characterize the hippocampal traveling theta wave (*Lubenov and Siapas, 2009*; *Patel et al., 2012*). Critically, however, previous work did not consider the impact of varying waveform asymmetry on the observed phase shifts. To address the asymmetry here, we used waveform-based methods of deriving estimates of theta phase using established methods (*Belluscio et al., 2012*; *Cole and Voytek, 2017*). We then computed the median time for specific identifiable portions of the waveform (peak, falling phase, trough, and rising phase) to shift 1 mm along the dorsal-ventral axis, and we report this as the conduction delay (see 'Materials and methods' for discussion of this quantity).

We found that the conduction delay was shortest for the falling phase (5.47 ms/mm [2.87–8.70]) and greatest for the peak to travel from dorsal sites to ventral sites (12.00 ms/mm [6.52–14.43]) as shown in *Figure 6*. These conduction delays differed significantly (rank sum [max]=0 [21], p=0.03, n=6). The conduction delay increased at a rate of 0.02 (ms/mm)/deg [0.017–0.025] between the falling phase and the peak, a relationship that was fit well by a linear trendline ($R^2$ = 0.82; *Figure 6C*). To compare the observed phase-dependent differences in conduction delay to the phase shift found using the Hilbert-transform-based approach described above, we converted these lags to degrees assuming an 8 Hz (125 ms period) theta rhythm. The converted phase shifts varied from 15.38°/mm [8.27–25.06] to 34.56°/mm [17.78–41.56] between the falling phase and peak phase, respectively. As expected given the phase smoothing that occurs during the filtering step of the Hilbert-based phase estimation approach, the phase shift estimate derived from the Hilbert-based approach (~26°/mm) falls between these limits (see dotted gray line on *Figure 6B*). These results show that theta is most synchronized during the falling phase and becomes progressively more desynchronized across phases until the peak of theta.

## Theta waveform changes are a source of phase shift along the dorsal-ventral axis of MEC

Because waveform asymmetries, even for an otherwise synchronized rhythm, would be sufficient to generate apparent phase offsets, we next analyzed how waveform changes along the dorsal-ventral axis may have contributed to the total conduction delay along the long axis of the MEC. To do so, the conduction delay between the first and each of the other channels that was attributable to varying asymmetry was calculated on the basis of the difference in the asymmetry index between these channels (see 'Materials and methods'). These calculations revealed that the conduction delay due to varying asymmetry was 3.56 ms/mm [5.89–1.56] (*Figure 7B and D*). This was significantly greater

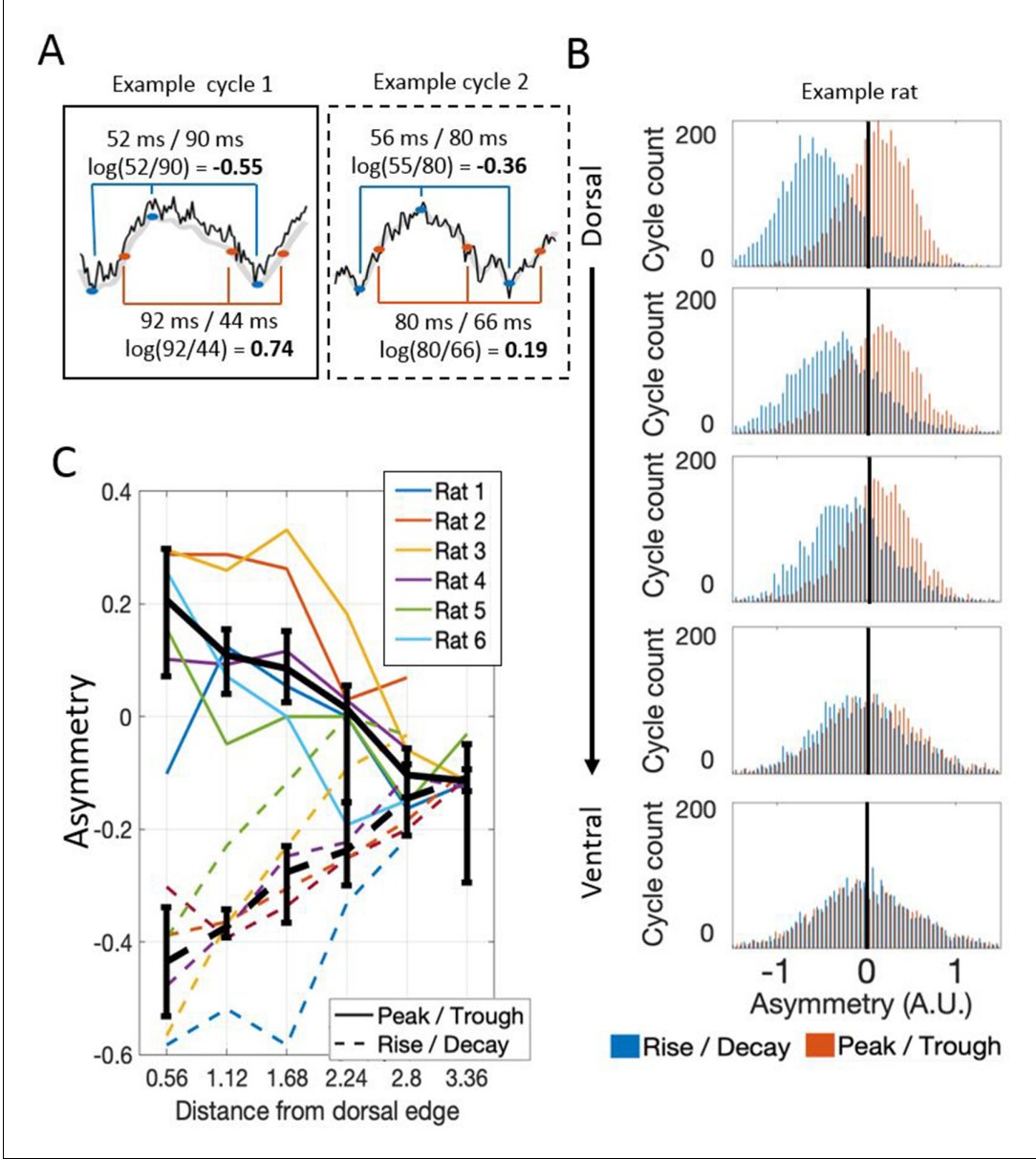

**Figure 5.** Theta waveform asymmetry varies along the dorsal-ventral axis of the MEC. (**A**) Schematic showing how waveform asymmetry was analyzed. Raw recordings (black) were filtered to 1–25 Hz (gray), theta troughs and peaks were identified as local minima and maxima, and rising and decaying phases were identified as the mid-points between the nearest local minima and maxima. The rise/decay asymmetry was calculated as the log of the ratio between the duration of the rising and decaying phases (marked with blue lines). The peak/trough asymmetry was calculated as the log of the ratio of the peak and trough durations (marked with red lines). (**B**) Representative example of rise/decay asymmetry and peak/trough asymmetry distributions observed over all theta cycles recorded across channels in a single trial from Rat 4. The histogram of rise/decay asymmetry values over theta cycles (blue bars) at dorsal sites was dominantly negative, indicating a fast rise and slow decay sawtooth waveform. Ventral sites were less negative, indicating more symmetric rise and decay times. The histogram of peak/trough asymmetry values over all theta cycles (orange bars) at dorsal sites was mostly positive, indicating that peaks were longer than troughs. Ventral sites were less positive, indicating more symmetric peak and trough durations. (**C**) Summary for all rats in the study. Colored lines show the median asymmetry values for individual rats. Black lines show the median across rats with 95% bootstrap confidence intervals on the within- subject effect of electrode position.

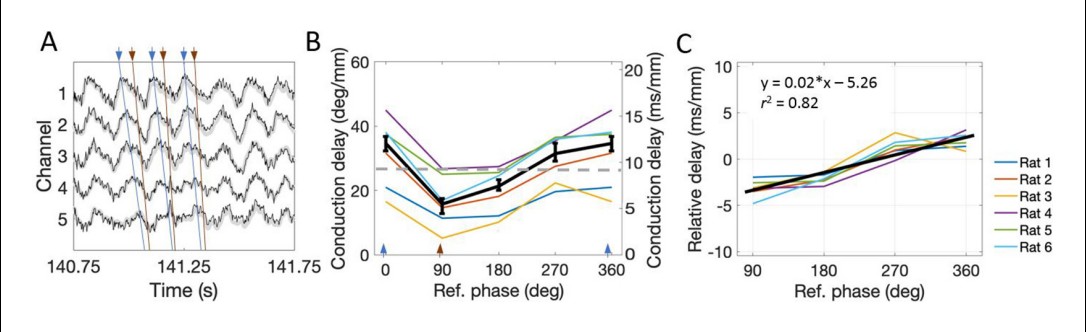

**Figure 6.** The conduction delay for theta to travel along the dorsal-ventral axis depends upon which phase is used as a reference point. (**A**) Representative example of LFP. Blue lines connect the peaks across channels and orange lines connect falling phases across channels. The orange lines appear steeper than the blue lines, indicating shorter conduction delays. (**B**) Across rats, the conduction delay is longest for theta peaks (0° or 360°, marked with blue arrowheads), shortest for falling phases (90°, marked with orange arrowhead), and increase progressively between 90° and 360°. Medians across rats are shown in black with 95% bootstrap confidence intervals on the within-rat effect of reference phase. (**C**) Subtracting the mean delay for each rat from the lines shown in panel (**B**) to focus on the within-rat effect of reference phase on delay shows a reliable positive correlation between reference phase and conduction delay. The black line shown the best-fit trendline ($R^2$ = 0.82) corresponding to the shown equation. Across panels, data from individual rats are shown as colored lines. Ref. = reference.

than zero (signed rank [max]=21 [21], p=0.03, n=6). Subtracting the asymmetry-related conduction delay (3.56 ms/mm [5.89–1.56]) from the total observed conduction delay (12.98 ms/mm [18.21–8.87]) left a conduction delay of 9.61 ms/mm [11.98–6.99] (**Figure 7C and D**). These data demonstrate that part of the total phase shift observed along the dorsal-ventral axis of the MEC can be attributed to theta waveform changes.

## Residual phase shifts after accounting for varying waveform asymmetry resemble those generated by weakly coupled oscillators

After accounting for the differences in waveform along the long axis of the MEC, substantial phase shift remained (see **Figure 7C**). This raised the question of what mechanism, beyond the waveform

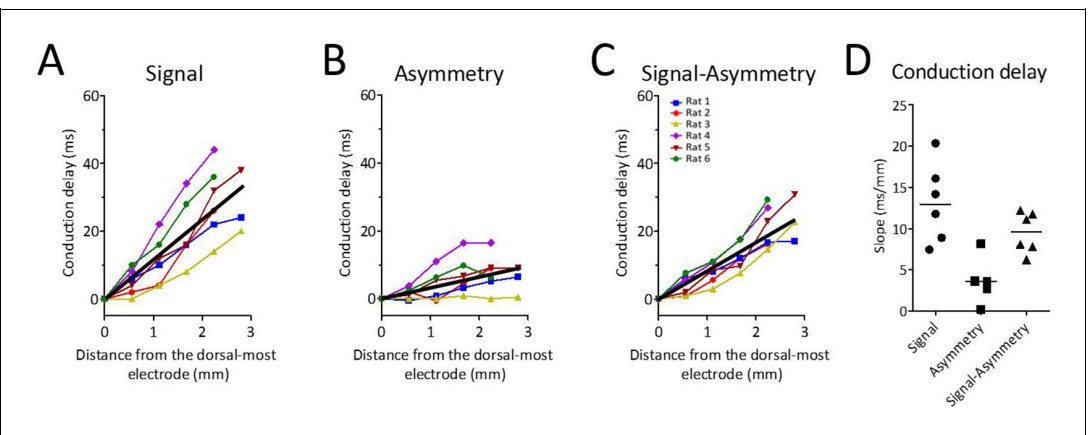

**Figure 7.** Theta waveform changes are a source of phase shift along the long axis of the MEC. (**A**) Observed phase shifts across channels (i.e., theta-phase-shifts) replotted from **Figure 2D** for ease of comparison. (**B**) Phase shifts expected given the changes in waveform asymmetry between sites (i.e., waveform-related-shifts). (**C**) Subtraction of waveform-related-shifts from theta-phase-shifts across electrode positions shows that the remaining phase shifts are relatively flat with respect to distance along the dorsal-ventral axis. (**D**) Comparison of the average phase shifts (degrees/mm) shown in panels (A–C) reveals significantly reduced phase shifts after subtracting waveform-related-shifts from the theta-phase-shifts.

asymmetry, could account for the apparent phase shifts. We sought to differentiate between two mechanisms in particular, weakly coupled-oscillators vs. fixed-delay propagation. These mechanisms make distinct, contrasting, predictions regarding how phase differences between electrodes should vary as a function of theta frequency. Because the delay in a fixed-delay propagation mechanism is fixed, the delays observed across electrodes (i.e. absolute delays) should not vary as a function of theta frequency. By contrast, normalizing the absolute delays to reflect what percentage of the cycle period the observed delay accounts for, in order to generate a 'relative delay,' should reveal an increasing relative delay as theta frequency increases and the period length decreases. This is illustrated in *Figure 8B*. Coupled-oscillators, on the other hand, exhibit absolute delays that change as a function of theta frequency changes. This happens in the coupled-oscillator model because changes in theta frequency occur across a set of oscillators that are hypothesized to be distributed along the dorsal-ventral axis of the MEC (e.g., resonant neurons receiving a common modulatory input). When these distributed oscillators receive input causing them to increase the frequency of their rhythmic activity, for example, the apparent delays between them will decrease. Accordingly, the observed delays (i.e., absolute delays) should decrease as theta frequency increases. However, because of the coupling, the relative phase alignments should remain largely the same. This causes the relative delays to be invariant across varying theta frequencies. This is illustrated in *Figure 8A*. To test between the fixed-delay and coupled-oscillator mechanisms, we examined the asymmetry-corrected data (see 'Materials and methods') to compare how cycle-by-cycle changes in theta frequency were related to either of two types of delays, which we refer to as absolute conduction delays and relative conduction delays.

The results revealed that the relationship between relative conduction delay (% theta period) and theta frequency (Hz) was not significantly different from zero (median slope 0.34% theta period/Hz [−0.42, 0.42]; signed rank [max]=12 [21], p=0.84, n=6; *Figure 8C*). By contrast, the relationship between the absolute conduction delay and theta frequency differed significantly from zero (median slope −3.12 ms/Hz [−3.52–2.71]; signed rank [max]=0 [21], p=0.03, n=6; *Figure 8D*). Comparing the correlation strength between the relative conduction delays (median rho = 0.02 [0.02–0.04]) and absolute conduction delays (median rho = 0.14 [0.09–0.19]) confirmed that conduction delay varied significantly more with theta frequency than did the phase difference (signed rank [max]=21 [21], p=0.03, n=6). These results are consistent with the pattern of effects that would be generated by a coupled-oscillator-based mechanism.

## Discussion

The goal of the work described here was to establish whether traveling waves in the entorhinal-hippocampal circuit are restricted to the hippocampus proper or whether the medial entorhinal cortex (MEC) exhibits traveling theta waves. We answered this question by recording at regular intervals along the dorsal-ventral axis of the MEC in freely behaving rats. These recordings showed reliable phase shifts like those described previously to exist along the dorsal-ventral (septal-temporal) axis of the hippocampus. We showed that varying waveform asymmetry contributed to the observed phase offsets and that the remaining phase offsets were consistent with those that would be generated by weakly coupled oscillators.

The present work adds the MEC to the growing list of circuits recognized to exhibit traveling waves when engaged (for a recent review see *Muller et al., 2018*). Importantly, however, it is also the first work to demonstrate that interacting functionally distinct circuits have matched traveling waves. This is a necessary condition for traveling waves to serve as a basic mechanism for coordinating interactions between distinct circuits. As such, the present work provides empirical support for existing theories regarding the utility of traveling waves in allowing the inherently distributed nervous system to operate as a coherent whole (*Ermentrout and Kleinfeld, 2001*; *Muller et al., 2018*).

Although the hippocampus and entorhinal cortex are neighboring anatomically, the traveling waves observed here are not volume conducted from the hippocampus for a couple of reasons. First, laminar profiles of theta in the MEC demonstrate that theta is generated locally given the existence of a local phase reversal spanning layer II (*Mitchell and Ranck, 1980*). Second, our analysis of the multi-unit activity distributed along the dorsal-ventral axis of the MEC revealed a phase-delayed sequence of activation that resembled that observed in the field potentials.

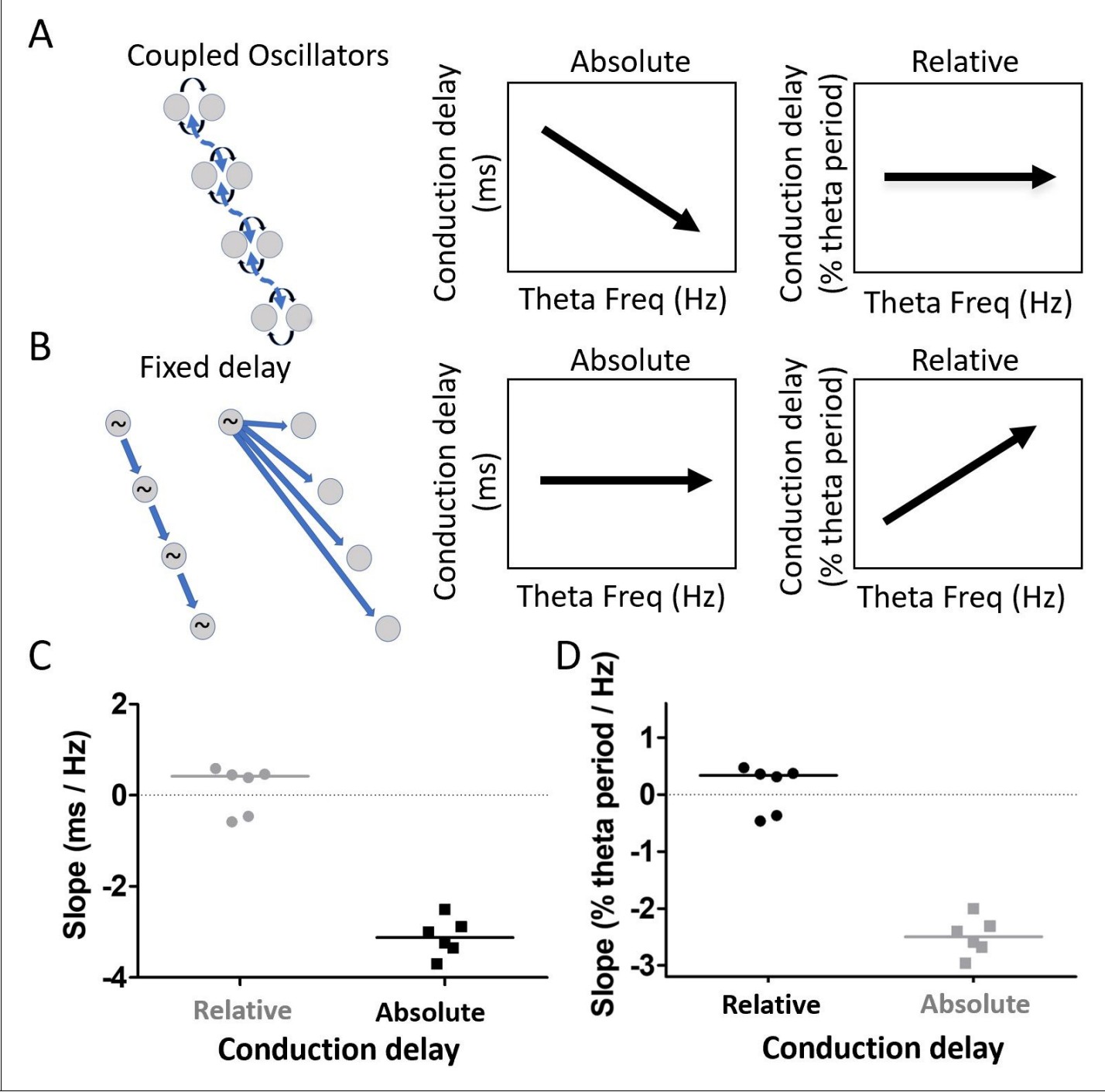

**Figure 8.** Residual phase shifts after accounting for varying waveform asymmetries are consistent with the pattern of effects that would be generated by a coupled-oscillator-based mechanism. (A) Schematic representation of a coupled oscillator mechanism for generating a traveling wave and the associated relationships between theta frequency and either absolute conduction delay or relative conduction delay. The phase locking between oscillators enforces a stable relative conduction delay across varying theta frequencies. These stable relative conduction delays correspond to decreasing absolute conduction delay as theta frequency increases. (B) Schematic representation of two fixed-delay mechanisms for generating a traveling wave and the associated relationships between theta frequency and either absolute conduction delay or relative conduction delay. Propagating excitatory pulses (left) and delayed excitations (right) can generate fixed-delay traveling waves. The fixed delay, by definition, creates fixed absolute conduction delays independent of theta frequency. These fixed delays, however, correspond to increasing relative conduction delays as theta frequency increases. (C) Slope relating relative conduction delay to theta frequency in terms of percent of period (%) per change in frequency (Hz) is shown in black. (D) Slope relating absolute conduction delay to theta frequency in terms of milliseconds (ms) per Hz is shown in black. To facilitate

*Figure 8 continued on next page*

*Figure 8 continued*

direct comparison of the slopes shown in panels (C) and (D), the data from each panel were converted on the basis of an assumed theta frequency of 8 Hz and plotted on the other panel in gray.

The existence of traveling theta waves in the MEC and hippocampus suggests the possibility of a wide-spread network of brain areas for which graded phase offsets regulate the functional dynamics. The phase gradients observed here, if extrapolated to span the 6–7 mm total length of the MEC would resemble those in the hippocampus. That is, if the ~26°/mm phase gradient that we observed continued along the reported 6–7 mm length of the MEC (*Insausti et al., 1997*; *Long et al., 2015*), the total theta phase shift along the MEC would be 158–185°. This is shorter than the ~200–220° phase shift that would be inferred if the 20.3–21.7°/mm phase gradient reported by *Lubenov and Siapas (2009)* continued across the ~10 mm length of the hippocampus. It more closely resembles the ~180° phase shift observed between the septal and temporal poles by *Patel et al. (2012)*. The MEC and hippocampus are only two of a broad set of brain areas that exhibit theta rhythmic activity (*Buzsáki, 2002*). If each of these areas had similar graded phase offsets, it would facilitate structured functional processing in this distributed circuit. Indeed, prior models of theta function have presupposed such a distributed set of graded phase offsets and have shown that such offsets can afford the related network with adaptive functional properties related to navigation and memory (e.g., *Blair et al., 2007*; *Burgess et al., 2007*; *Dickson et al., 2000*; *Hasselmo et al., 2009*).

Crucial for linking function to physiology is the question of how this traveling wave is generated. We found that a portion of the phase shifts observed across the MEC were related to differences in theta waveform. At dorsal sites, a sawtooth pattern was observed, characterized by a rapid rising phase and a gradual descending phase, whereas at ventral sites, a more sinusoidal pattern was observed. Oscillation waveforms reflect the properties of their underlying physiological generators (for a review, see *Cole and Voytek, 2017*). The waveform changes that we observed across the MEC indicates that the underlying generators vary along the dorsal-ventral axis.

We hypothesize that the rapid rise of the sawtooth theta pattern observed at dorsal sites reflects an active local current source generated by the recruitment of local fast-spiking feedback inhibition (*Pastoll et al., 2013*). The same feedback inhibition may functionally support the formation of grid cell attractor networks (*Pastoll et al., 2013*). Notably, a prominent type of fast spiking feedback inhibitory neuron, parvalbumin expressing (PV+) interneurons, are distributed across the MEC in a gradient with high densities at dorsal sites and low densities at ventral sites (*Beed et al., 2013*). This distribution matches the anatomical distribution of the saw-tooth waveform pattern observed here and could account for the observed waveform changes.

Alternatively, the waveform shifts could be related h-channels, which underlie theta generation (*Kocsis and Li, 2004*) in the medial septum and have graded strength along the dorsal-ventral axis of the MEC (*Giocomo and Hasselmo, 2009*). H-currents, too, are associated with the generation of grid tuning (*Giocomo et al., 2007*; *Giocomo et al., 2011*). Whether inhibitory tone or h-currents, the fact that the same mechanism may underlie generation of the MEC traveling wave and grid cell attractor networks implies a deep functional connection between the traveling wave and spatial processing in the MEC.

In summary, our data demonstrate the existence of theta traveling waves in the medial entorhinal cortex that are matched to the traveling waves known to exist in hippocampus proper. The existence of matched traveling waves in interacting cortical areas supports the hypothesis that traveling waves are a basic organizing mechanism for coordinating cortical information processing. With regard to the underlying mechanism, our findings that theta waveform varied in a graded fashion across the axis indicate that the traveling wave is not generated by a propagating wave front but rather by varying physiological generators across the axis, constraining mechanistic models of theta generation and function in the entorhinal-hippocampal circuit.

## Materials and methods

All experiments were performed in accordance with the National Institute of Health guide for the care and use of laboratory animals (NIH Publications No. 80–23) and were approved by the Bloomington Institutional Animal Care and Use Committee.

## Animals

Recordings were made in six male Long-Evans rats. All animals weighed 350–500 g at the time of surgery, were individually housed, and were maintained at 90% of their free-feeding weight following their full recovery after surgery. Animals were maintained on a 12:12 hour light-dark cycle, all procedures were conducted during the light cycle.

## Behavioral protocol

All recordings were performed as rats completed laps of a circular track in order to collect pieces of sweet cereal in 25-min-long testing trials. The track was made from 8-cm-wide black acrylic with 5-cm-tall opaque black plastic walls on both the inside and outside edges (*Figure 1A*). The circle had a diameter of 112 cm and was elevated ~100 cm from the floor by six legs. Animals were rewarded for each complete lap run in either direction. The rewards were delivered in a fixed position by an experimenter standing at arm's length from the track. Numerous landmarks (e.g., experimenter, desk, door, pedestal, shelving) were visible from the track in the recording room. Rats received at least two weeks of training, consisting of one or more 20 min trials each day, prior to the key recording trials.

## Electrode array construction

Custom multi-shank electrode arrays were built to record at regular intervals along the dorsal-ventral axis of the MEC. The positions of individual electrodes were fixed with respect to all other electrodes in the array. To build the electro array, 15–16 silica tubes (150 μm in diameter) were arranged and glued together in the horizontal plane. Individual shanks were created by loading stereotrodes or tetrodes (polytrodes) into single silica tubes or by merging the polytrodes loaded into adjacent silica tubes. Polytrodes were constructed by twisting two or four 12 μm or 25 μm diameter nichrome wires (A-M Systems) together. In one array, each shank was comprised of a single polytrode. Four arrays were built to have two polytrodes per shank so as to have a vertical offset between the recording tips of the polytrodes within a shank of ~280 μm. This design permitted us to use the deeper of the two as a scout electrode, seeking out the phase reversal that exists between layers II and I (*Mitchell and Ranck, 1980*). With this tip in layer I, the relatively shallow electrode was reliably positioned in layers II–III. The probe with up to six stereotrodes per shank and a vertical spacing of 450 μm was used to sample across the layers of the MEC more extensively (*Figure 3A and B*).

Separate shanks were created in every other silica tube, resulting in a 300 μm separation between adjacent shanks. The total length of each successive shank from the silica increased by 470 μm. These spacings were selected so that the electrodes would track the ~57° orientation of the MEC relative to the horizontal plane (*Figure 1B*). The net distance between the tips of adjacent shanks was net ~566 μm. The full array of eight shanks spanned ~3.9 mm in the dorsal-ventral axis of MEC.

Precise vertical spacing of stereotrodes within and between shanks was accomplished through the use of a micrometer graduated to a scale of 10 μm mounted to a microslicer (Stoelting part # 51425). In one array where neighboring shanks were glued together to create half as many shanks, the net resulting spacing was measured through inspection of the array under a dissection microscope equipped with a reticle (Amscope part # EP10 × 30R). Assembled arrays were connected to an electrode interface board and anchored to a single hand-built microdrive (*Vandecasteele et al., 2012*). Finally, electrode tips were gold plated with a golden solution diluted 90% with a solution of 1 mg/mL polyethelene glycol in distilled water to bring the impedance at 1 kHz down to ~250 kΩ.

## Electrode array implantation and positioning

For electrode array implantation, rats were anesthetized with isoflurane, and a local anesthetic (lidocaine 2 mg/kg and bupivicaine 1 mg/kg) was administered subcutaneously to the scalp before any surgical incision. After the scalp was gently retracted and the surface of the skull was exposed, four or five anchor screws and one ground screw (located over the central cerebellum) were affixed to the skull. A craniotomy was performed over the right MEC, at 4.5 mm lateral and extending from the transverse sinus to ~2.5 mm anterior of the sinus. The electrode array was aligned 0.3 mm anterior to the transverse sinus and lowered 4.5–5 mm into the brain at the time of surgery. The microdrive was attached to the screws and the skull with dental acrylic. A copper mesh was then used to

fashion a protective cage around the drive and solidified using dental acrylic (*Vandecasteele et al., 2012*).

The rats recovered for seven days after surgery before behavioral testing and recording began. The electrode array was then stepped down to the MEC by advancing the microdrive in 140 µm increments until theta activity was prominent over most electrodes of the array (typically 5.7–6.0 mm at the deepest electrode). The final placement was established when the ventral-most recording sites on each shank crossed the boundary between layers II and I, as indicated by a reversal of theta phase between the polytrodes of a single shank, leaving the dorsal-most sites on each shank in layer III (*Mitchell and Ranck, 1980*). For recordings of multi-unit activity in layer II, the microdrive was adjusted between 50–200 µm, in ~50 µm intervals, until prominent high-frequency multi-unit activity was visible (e.g., *Figure 4A*).

## Electrophysiological recordings

Recordings were performed using a 32-channel headstage (Intan Technologies part RHD2132) connected directly to a USB interface board (Intan Technologies part an RHD2000). Data acquisition was controlled via a variant of the OpenEphys software platform customized to allow sync-pulses generated by a thresholded version of the electrophysiology to be sent to the behavioral tracking camera system. Field potential recordings were referenced to ground and written to disk at 30 kHz sampling rate.

## Histology

At the end of the experiments, the rats were deeply anesthetized with isoflurane and intracardially perfused with buffered formaldehyde solution (4%). Then, the brains were removed from the skull and soaked in 30% sucrose solution until saturated. The brains were then sectioned in the sagittal plane into 40 µm slices. Slices were mounted to slides and stained with cresyl violet to identify electrode tracks. Electrode position at the time of recording was established by cross-referencing images of these slides to logs of electrode movements, allowing for corrections when electrodes were advanced further after the recording. Raw histological micrographs are shown as *Figure 1— figure supplement 1*.

## Data analysis

Analyses were performed in MATLAB using custom scripts and with the CMBHOME toolbox.

### Electrode selection criteria

Which specific electrodes were used in the analyses was determined in a multistep selection process. First, any electrodes with unstable signal, either because it was flat or because the impedance was abnormal (<100 kΩ or >8 kΩ at 1000 Hz), were excluded from consideration. Second, the position of the array relative to the layers of the MEC was estimated by cross-referencing the known probe geometry to the observed electrophysiological markers. With regard to electrophysiological markers, specific attention was paid to theta power, theta phase reversals and, when available, the phase locking of local spiking activity. Given these estimates, the polytrode positioned closest to layer III from each shank was selected. Finally, with these stereotrodes identified, one of the two electrodes from each polytrode was selected. The final selections were reviewed for accuracy with histological verification of the electrode pathway.

### Data inclusion criteria

Analyses were performed on epochs of data that were free of artefacts and had prominent theta fluctuations unless otherwise noted. Artefacts were defined as voltage swings greater than five times larger than the root mean square (RMS) of the signal. Identified artefacts, along with a 500 ms buffer before and after the artefact, were marked to be excluded. Because artefacts inflate the RMS of the signal, potentially creating inappropriately high thresholds for recognizing artefacts, the artefact removal procedure was applied to the resulting 'cleaned' data repeatedly until no further artefacts were found. For electrodes with acceptable impedance, this usually resulted in <5% of the data being removed. From the remaining data, epochs with high-amplitude theta were selected for further analysis. High-amplitude theta was defined as that when the amplitude of theta (extracted by a

zero phase-lag fourth-order bandpass Butterworth filter with 6 and 12 Hz cutoff frequencies) was greater than 400 µV for 500 ms or longer. Any theta amplitude dips below this threshold that lasted for 100 ms or less were ignored.

### Theta phase estimation

Two approaches were taken to estimate theta phase. The first used the Hilbert transform method, matching the approach used previously (*Lubenov and Siapas, 2009*; *Patel et al., 2012*; *Zhang and Jacobs, 2015*). That is, raw traces were filtered to the 6–12 Hz band using a zero phase-lag fourth-order bandpass Butterworth filter. The Hilbert transform was then used to convert the filtered signals into the complex plane and the angle from the real axis was taken as an estimate of the theta phase. The second approach uses the 'waveform method' previously established to estimate the phase of non-sinusoidal signals (*Belluscio et al., 2012*; *Cole and Voytek, 2017*). That is, raw traces were filtered to the 1–25 Hz range using a zero phase-lag fourth-order bandpass Butterworth filter, local maxima and minima separated by at least 83 ms and no more than 250 ms were identified and marked as the 0° and 180° phases, respectively. For analyses requiring localization of the 90 ° and 270 ° phases, these were localized to the time point at which the filtered voltage was closest to the mean of the nearest minima and maxima.

### Cycle-triggered averages theta waves

To establish a picture of the average alignment of theta across the dorsal-ventral axis, we computed event related averages of the raw LFP triggered on the peak of theta on the dorsal-most channel. Importantly, this procedure did not use filtered signals, thereby avoiding the possibility that variations in theta waveform were altered. The peak of theta on the dorsal-most channel was accomplished using the waveform-based estimation of theta phase (*Belluscio et al., 2012*; *Cole and Voytek, 2017*).

### Phase locking

To estimate the reliability of the phase offset between electrodes, we estimated the theta phase using the Hilbert transform method and then calculated the mean resultant length of the phase differences between electrodes. The phase differences themselves were computed as the circular difference between the instantaneous theta phase estimates from each of the channels.

### Theta band coherence

Theta band coherence was calculated as the average coherence across the 6–12 Hz frequency band (1 Hz increments). The coherence values themselves were derived by pass the artefact-free data to the Matlab function cohere.m. Importantly, this analysis was not restricted to epochs of high theta amplitude.

### Conduction delays of specific theta phases

Conduction delay refers here to the time lag for a signal to appear on a relatively ventral site after appearing on the dorsal-most site. The label conduction delay is not intended to imply literal movement or propagation and is intended to be agnostic about mechanism. Conduction delays were computed separately for each of four distinct phases of the theta signal. These were the troughs, rising phases, peaks, and falling phases as identified with the waveform-based phase estimation approach. For a given phase (e.g., peak), all timepoints at which the phase of the dorsal-most channel was in that phase were taken as individual cycles. For each of those cycles, the timepoints when the remaining channels were also in that same phase that were most proximal (either forward or backward in time) to the timestamp of the dorsal-most site were identified. Importantly, this did not introduce bias with respect to propagation direction. For each cycle, we then performed a robust regression (via Matlab function robustfit.m) between the estimated anatomical position of each channel along the dorsal-ventral axis and the temporal offset of each timestamp relative to the reference timestamp on the dorsal-most channel. The resulting slope (ms/mm) was taken as an estimate of the conduction delay for that cycle. Cycles for which the regression failed to converge were dropped from the analysis. Between 3500 to 5000 individual theta cycles were analyzed for each phase in

each rat. The median over all of these cycles was taken as the conduction delay for that phase for a given rat.

### Analysis of multiunit spiking activity

To test whether neuronal activity also exhibited traveling waves, the electrodes were pushed deeper into layer II, as indicated by strong trough-locked multi-unit activity. Multi-unit activity was isolated from the broadband signal by bandpass filtering the raw LFP (zero phase-lag fourth-order Butterworth bandpass filter with 600 and 3000 Hz cutoff frequencies). Then, to extract the theta-timescale modulations of this spiking band, we bandpass filtered the amplitude envelope of the resulting spiking-band activity to the 4–12 Hz band (zero phase-lag third-order Butterworth filter). To test for systematic lags in the change of this theta modulated multi-unit activity, we computed the temporal cross-correlogram between each channel and the dorsal-most channel and, for each, extracted the lag to the maxima within a window of +/– 120 ms. Finally, we performed a regression between the extracted lags to the electrode positions to determine the average conduction delay in ms/mm. To compare the obtained multi-unit conduction delay to that expected from the field potential, we compared the multi-unit regression slope to the one derived from analyzing the LFP theta from same subset of three rats from which the multi-unit activity was recorded.

### Asymmetry analysis

To test for changes in the shape of the theta waveform, we used the asymmetry index described by *Cole and Voytek (2019)*. Briefly, theta peaks and trough were identified using the waveform-based method described above. The waveform asymmetry was evaluated on the basis of the rise-decay symmetry bases, the ratio of the rising phase duration to the decaying (falling) phase duration. The $\log_{10}$ transform of this ratio was then taken to create the index. This asymmetry index was computed for each theta cycle. The median across all theta cycles was then used to reflect the waveform asymmetry for that channel for that recording.

### Asymmetry-corrected data

Because asymmetry differences across electrodes alone would generate apparent conduction delays, we derived cycle-by-cycle conduction delay 'corrections' on the basis of the observed asymmetries. To apply this correction, we first calculated the conduction delay that could be expected by the asymmetry alone (i.e., the asymmetry-related delay) for every theta cycle. The asymmetry-related delay for each cycle was computed according to the following formula:

$$Delay_{Asym} = (D_{falling}^{elec^i} - D_{rising}^{elec^i}) - (D_{falling}^{elec^j} - D_{rising}^{elec^j})$$

where $D_{falling}$ and $D_{rising}$ were the durations of the falling and rising phases, respectively. This formula yields the difference in the asymmetry between two electrodes on a given theta cycle in milliseconds. The asymmetry-corrected conduction delays were computed by subtracting this asymmetry-related delay from the delays derived from analyzing the theta phases. The peak phases were used as the key reference phase, in particular because this was the phase at which the greatest total conduction delays were observed.

### Comparison to coupled-oscillators and fixed-delay models of traveling waves

To test whether the traveling waves that remained after accounting for variable waveform asymmetries were most consistent with a model of traveling wave generation based on weakly coupled oscillators or fixed-delay based mechanisms (e.g., propagating excitation or variable delays of a single oscillator), we compared how cycle-by-cycle changes in theta frequency were related to cycle-by-cycle changes in *absolute* conduction delay versus cycle-by-cycle changes in *relative* conduction delay. Absolute conduction delay was computed as described above using theta peaks as the key reference phase after the asymmetry related component was subtracted (see above). Relative conduction delay is the absolute conduction delay for a given cycle divided by the period length of that cycle. For both, we asked whether there was a reliable correlation with variance in the cycle-by-cycle theta frequency (computed as the inverse of the period). Traveling waves generated by weakly

coupled oscillators should have no significant correlation between theta frequency and relative conduction delay but should have a negative correlation between theta frequency and absolute conduction delay. Traveling waves generated by fixed-delay mechanisms should have a positive correlation between theta frequency and relative conduction delay but no significant correlation between theta frequency and absolute conduction delay (*Ermentrout and Kleinfeld, 2001*).

## Statistics

Because the changes were reproduced in each animal (*Figure 2*), a sample size of six animals was chosen. This is consistent with previous electrophysiological studies of LFP using similar sample sizes (*Lubenov and Siapas, 2009*). Summary statistics are reported as medians with 95% bootstrap confidence intervals, based on 500 permutations, in square brackets. Nonparametric statistical testing of differences was done throughout. In the case of nonpaired tests, Wilcoxon rank-sum tests were performed, In the case of paired tests, Wilcoxon signed-rank tests were performed. The only parametric testing done was when analyzing correlation, using Pearson correlation, so as to assess the goodness of fit of coincidently performed linear regressions.

## Acknowledgements

We thank the Whitehall Foundation, CONACYT (232364), and Indiana University for their generous support of this project. We are also grateful to J Hinman, M Brandon, S McKenzie, M E Olvera-Cortés, and J M Cervantes-Alfaro for their comments and suggestions. Finally, we thank the staff of Indiana University laboratory animal resources for their care and attention to the subjects of this work.

## Additional information

### Funding

| Funder | Grant reference number | Author |
|---|---|---|
| Whitehall Foundation | | Ehren L Newman |
| Consejo Nacional de Ciencia y Tecnología | 232364 | Jesus J Hernández-Pérez |
| Indiana University | | Ehren L Newman |

The funders provided resources for the study design, data collection, and interpretation.

### Author contributions

J Jesús Hernández-Pérez, Software, Formal analysis, Supervision, Funding acquisition, Validation, Investigation, Visualization, Methodology, Writing - original draft, Writing - review and editing; Keiland W Cooper, Formal analysis; Ehren L Newman, Conceptualization, Resources, Software, Formal analysis, Supervision, Funding acquisition, Validation, Investigation, Visualization, Methodology, Writing - original draft, Project administration, Writing - review and editing

### Author ORCIDs

J Jesús Hernández-Pérez https://orcid.org/0000-0001-6963-4712
Keiland W Cooper https://orcid.org/0000-0002-0358-9645
Ehren L Newman https://orcid.org/0000-0001-7006-4112

### Ethics

Animal experimentation: This study was performed in strict accordance with the recommendations in the Guide for the Care and Use of Laboratory Animals of the National Institutes of Health. All of the animals were handled according to approved institutional animal care and use committee (IACUC) protocols (#18-026) of Indiana University, Bloomington. The protocol was approved by the Committee on the Ethics of Animal Experiments of Indiana University, Bloomington (Animal Welfare

Assurance Number D16-00587). All surgery was performed under isoflurane anesthesia, and every effort was made to minimize suffering.

### Decision letter and Author response
Decision letter https://doi.org/10.7554/eLife.52289.sa1
Author response https://doi.org/10.7554/eLife.52289.sa2

## Additional files
### Supplementary files
• Transparent reporting form

### Data availability
Data are available on CRCNS (http://crcns.org/data-sets/hc/hc-27/about-hc-27) under the doi (http://dx.doi.org/10.6080/K0C53J2R). Users must first create a free account (https://crcns.org/register) before they can download the datasets from the site.

The following dataset was generated:

| Author(s) | Year | Dataset title | Dataset URL | Database and Identifier |
|---|---|---|---|---|
| Hernández-Pérez JJ, Cooper KW, Newman EL | 2020 | Extracellular recordings from across the dorsoventral axis of the medial entorhinal cortex of the rat | http://crcns.org/data-sets/hc/hc-27 | Collaborative Research in Computational Neuroscience, 10.6080/K0C53J2R |

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
