## [Decision Letter]

**Acceptance summary:**

This paper provides evidence that activity in medial entorhinal cortex forms a traveling wave in the theta band that travels along the dorso-ventral direction. This is evidenced in local field potential and single unit activity, resembling the traveling wave reported in hippocampus proper and with important implications for the communication between both regions.

**Decision letter after peer review:**

Thank you for submitting your article "Medial entorhinal cortex activates in a traveling wave" for consideration by *eLife*. Your article has been reviewed by three peer reviewers, one of whom is a member of our Board of Reviewing Editors, and the evaluation has been overseen by Laura Colgin as the Senior Editor. The reviewers have opted to remain anonymous.

The reviewers have discussed the reviews with one another and the Reviewing Editor has drafted this decision to help you prepare a revised submission.

Summary:

This manuscript reports for the first time the propagation of theta waves along the dorso-ventral axis of the MEC. Using electrode arrays to sample a range of dorso-ventral and laminar positions of MEC, the authors observe a gradual phase shift along the dorso-ventral axis for theta oscillations on a rate and scale similar to the septo-temporal phase shifts reported in the hippocampus. The theta phase shifts are observed on both theta range-filtered LFP and multiunit activity, and do not originate from differences in electrode laminar position. In addition, the authors analyze the mechanisms underlying the theta wave propagation. One contributing factor is a change in the shape of theta waves from more asymmetric in dorsal MEC to more symmetric in ventral MEC. After subtracting this effect, the remaining phase shifts scale with the size of the theta waves, consistent with the prediction of weakly-coupled oscillator models and at odd with models stipulating fixed delays.

All three reviewers agreed that this paper represents a potentially interesting and important finding, but there are also some potentially significant issues to address, as listed below.

Essential revisions:

1) The key methodological problem here is to separate the phase and waveform shifts between Superficial-Deep layers from those between equivalent Dorso-Ventral locations. In one animal there are multiple shanks with contacts at multiple depths along them in which both the S-D phase shifts and D-V phase shifts can be measured and the authors can try to control for the potentially confounding effect of one on the other (Figure 3).

In another approach the array is moved to record multi-unit activity in layer II (please clarify if this is a single animal or all 6, and whether the array is moved in each from the previous location near layer III) and the analysis performed on the total spiking rather than the LFP.

For the animals used in the standard LFP recording protocol (5 animals?) this cannot be done. Each shank has contacts at two depths, the first can be used to check that it has gone through the theta phase inversion between layers I and II – leaving the second hopefully in layer III. However, it is not clear that a consistent S-D location can be achieved across D-V locations with a single fixed array, especially since the layers are not perfect flat planes whereas the array seems to be linear (is this right?). The authors could use the level at which each first contact passed the theta phase inversion to make an estimate for each shank of the location of the second contact relative to layer III, using histology to determine the angle of the shank relative to the layer. This could then be regressed against measure of phase to see if it could be a confound (and possibly to remove this confound).

In summary, it is not clear whether the authors could be sure that their significant results (in multiple animals) reflected D-V location rather than S-D location. This has to be clarified.

2) The authors argue that the MEC traveling wave is not volume conducted from the hippocampus, which may be true, but it's hard to buy the story that the traveling wave in MEC is independent, or at a different speeds from the traveling wave in the hippocampus given the very strong reciprocal connections between these structures. They estimate that, based on the% of the DV axis covered in the MEC, that a similar coverage of the hippocampus would yield a 50 degree shift, whereas they see a >70 degree shift. How confident can the authors be about this estimation? This is not just about volume conduction – this has major implications about how the hippocampus and MEC communicate. If it’s true that the MEC and HPC have traveling waves at different speeds then it means that MEC inputs to the hippocampus are not arriving at a consistent phase of hippocampal theta. This would have major implications for how the hippocampus processes MEC inputs and I think the authors should reflect on how confident they are about this before making this very bold claim. If they are worried about volume conduction they should analyze their units – if they want to make this very bold claim about phase differences between MEC and hippocampus they should perform dual-recordings.

---

## [Author Response]

Essential revisions:1) The key methodological problem here is to separate the phase and waveform shifts between Superficial-Deep layers from those between equivalent Dorso-Ventral locations. In one animal there are multiple shanks with contacts at multiple depths along them in which both the S-D phase shifts and D-V phase shifts can be measured and the authors can try to control for the potentially confounding effect of one on the other (Figure 3).In another approach the array is moved to record multi-unit activity in layer II (please clarify if this is a single animal or all 6, and whether the array is moved in each from the previous location near layer III) and the analysis performed on the total spiking rather than the LFP.For the animals used in the standard LFP recording protocol (5 animals?) this cannot be done. Each shank has contacts at two depths, the first can be used to check that it has gone through the theta phase inversion between layers I and II – leaving the second hopefully in layer III. However, it is not clear that a consistent S-D location can be achieved across D-V locations with a single fixed array, especially since the layers are not perfect flat planes whereas the array seems to be linear (is this right?). The authors could use the level at which each first contact passed the theta phase inversion to make an estimate for each shank of the location of the second contact relative to layer III, using histology to determine the angle of the shank relative to the layer. This could then be regressed against measure of phase to see if it could be a confound (and possibly to remove this confound).In summary, it is not clear whether the authors could be sure that their significant results (in multiple animals) reflected D-V location rather than S-D location. This has to be clarified.

The question of whether differential placement of electrodes *across the MEC layers* (S-D axis named here) contributed to the phase shifts that we attribute to the positioning *along the MEC layers* (D-V axis named here) is a good one. This is important to rule out. A number of lines of evidence rule this out as a major contributor to our results.

1) The strongest line of evidence comes from the extensive body of literature that describes theta phase distributions across the MEC layers (Mitchell and Ranck, 1980; Alonso and García-Austt, 1987; Chrobak and Buzsáki, 1998; Mizuseki et al., 2009; Quilichini et al., 2010). This literature has consistently demonstrated that 1) theta phase is the same across MEC layers III-V, and 2) that theta phase reverses between layers III and I with a zone of null theta amplitude between them. We acknowledge that there is some phase instability as electrodes approach the phase reversal between EC3 and EC1. Critically, however, this instability is accompanied by a drop and eventual loss of theta power. Analysis of theta power across our electrodes revealed no systematic differences in power as a function of position along the MEC layers (D-V axis). The results of this analysis have been added to manuscript Figure 2 as panel G. The lack of systematic differences in theta power across recording sites indicates that our electrodes do not vary in their positioning relative to the phase reversal zone. Based on this, it is safe to conclude that the systematic phase shifts that we do observe are not driven by phase instabilities near the phase reversal. Because there are no other phase shifts across MEC layers, it is also safe to conclude that no across layer phase differences confounded our results.

2) The second line of evidence is our data examining the distribution of phase shifts across and along the MEC layers simultaneously in one animal, shown in Figure 3. This analysis replicates the reported lack of phase shifts across MEC layers (~S-D axis) in the five published works cited in point 1 but also extends the existing work by simultaneously showing the existence of phase shifts along the MEC layers (~D-V axis). We acknowledge that this is but one animal and thus no statistical testing is possible. As noted in the previous point, prior published reports rule out the ‘across layer’/S-D axis phase shift confound. The reason we none-the-less include Figure 3 and the associated description in the results is two-fold: 1) It enables us to share data that we have illustrating the phase gradients across and along the layers which we believe helps make the overall understanding of our main result clearer; and 2) We believe it is more compelling to see some direct empirical evidence of the lack of contribution of across layer placements rather than none in the case that we relinquished our handling of this point to a string of citations in the Introduction and Discussion.

3) The third line of evidence, as recognized by the reviewers, is the multiunit activity which shows the qualitatively similar pattern of sequential activation along the length of the MEC.

These three points should leave no doubt that the phase shifts reported here can be safely attributed to electrode positioning alongthe MEC layers and not *across* the MEC layers.

We were appreciative for the reviewers’ suggested approach to estimating and handling variability in the precise placement of electrodes relative to the cell layers. Indeed, our conviction that this analysis would be valuable analytical tool in pursuing this line work, we dedicated considerable effort to implementing the suggest analysis. Ultimately, however, between the granularity of the movements of the electrodes (typically 50-100 μm) and the graded nature of the physiological shifts across layers (e.g., theta power and multi-unit activity), we were not able to improve our estimates of the S-D positioning of the electrodes over the approximations we had from knowing the probe design. Never the less, for the sake of satisfying the curiosity of the reviewers, we include Author response image 1 with the results we obtained when applying the approach and analysis to the 5 animals that had any variability in the across-layer dimension as well as the along-layer dimension. Please note, though the results shown in Author response image 1 are consistent with the data shown in the current Figure 3, these results should not be included in the manuscript or supplemental materials because the limited sampling across layers in at least the bottom three of the rats restricted the variability in that dimension and thereby generated highly unreliable phase gradient estimates. The lack of variability is visible in the figure in that the values along the x-axis are highly clustered in the right-most column of panels.

Regarding the reviewers’ questions about the number of animals included in the analysis of traveling waves of multi-unit activity (Figure 4): We have ensured that the figure caption and the corresponding portion of the results clearly state that three animals were included in this analysis and that the electrodes were moved from layer III to enable this analysis.

**Author response image 1. respfig1:** Phase shifts are reliably positive and strongly associated with electrode position along the MEC layers and unreliably / not related to positioning across the MEC layers. (**A**) Reconstructed electrode positioning in the MEC based on analysis of physiological markers observed at the time of recording and by comparing the changes in physiological markers and electrode movements over days. (**B**) Phase differences between pairs of electrodes are plotted as a function of the spatial offset of the electrodes between recording sites along the MEC layers. (**C**) Same as B but phase offsets are plotted as a function of the spatial offset across the MEC layers. Across these five rats (shown across the rows), a consistent positive relationship is observed when phase offsets are plotted as a function of the spatial offsets along the MEC layer but not when plotted as a function of the spatial offset across the MEC layers.

2) The authors argue that the MEC traveling wave is not volume conducted from the hippocampus, which may be true, but it's hard to buy the story that the traveling wave in MEC is independent, or at a different speeds from the traveling wave in the hippocampus given the very strong reciprocal connections between these structures. They estimate that, based on the% of the DV axis covered in the MEC, that a similar coverage of the hippocampus would yield a 50 degree shift, whereas they see a >70 degree shift. How confident can the authors be about this estimation? This is not just about volume conduction – this has major implications about how the hippocampus and MEC communicate. If it’s true that the MEC and HPC have traveling waves at different speeds then it means that MEC inputs to the hippocampus are not arriving at a consistent phase of hippocampal theta. This would have major implications for how the hippocampus processes MEC inputs and I think the authors should reflect on how confident they are about this before making this very bold claim. If they are worried about volume conduction they should analyze their units – if they want to make this very bold claim about phase differences between MEC and hippocampus they should perform dual-recordings.

We believe that this concern is driven by a misunderstanding of the argument we had tried to make. It was not our intention to say that the areas would have different phase shifts over comparable% of the DV axis. We are in full agreement that this would indeed be troubling / consequential if it were the case. We considered trying to clarify the discussion text to better communicate what meant. However, this would have risked further confusion about what is ultimately a minor point. Given that our goal was simply to argue against the risk of volume conduction, we decided it was easier to replace this argument with reference to the unit activity as suggested by the reviewers. We are grateful that the reviewers identified the risk of misunderstanding of this part of the manuscript so that we could remove it.